# Experimental and Computational Investigation of the Oxime Bond Stereochemistry in c-Jun N-terminal Kinase 3 Inhibitors 11*H*-Indeno[1,2-*b*]quinoxalin-11-one Oxime and Tryptanthrin-6-oxime

**DOI:** 10.3390/pharmaceutics15071802

**Published:** 2023-06-23

**Authors:** Vladislava V. Matveevskaya, Dmitry I. Pavlov, Anastasia R. Kovrizhina, Taisiya S. Sukhikh, Evgeniy H. Sadykov, Pavel V. Dorovatovskii, Vladimir A. Lazarenko, Andrei I. Khlebnikov, Andrei S. Potapov

**Affiliations:** 1Nikolaev Institute of Inorganic Chemistry, Siberian Branch of the Russian Academy of Sciences, 3 Lavrentiev Ave., 630090 Novosibirsk, Russia; matveevskaya@niic.nsc.ru (V.V.M.); pavlov@niic.nsc.ru (D.I.P.); sukhikh@niic.nsc.ru (T.S.S.); sadykov@niic.nsc.ru (E.H.S.); 2Kizhner Research Center, National Research Tomsk Polytechnic University, 30 Lenin Ave., 634050 Tomsk, Russia; ark4@tpu.ru; 3National Research Centre “Kurchatov Institute”, Kurchatov Square 1, 123182 Moscow, Russia; paulgemini@mail.ru (P.V.D.); vladimir.a.lazarenko@gmail.com (V.A.L.)

**Keywords:** kinase inhibitor, oxime, crystal structure, nuclear magnetic resonance, rotation barrier

## Abstract

11*H*-Indeno[1,2-*b*]quinoxalin-11-one oxime (**IQ-1**) and tryptanthrin-6-oxime are potent c-Jun N-terminal kinase 3 (JNK-3) inhibitors demonstrating neuroprotective, anti-inflammatory and anti-arthritic activity. However, the stereochemical configuration of the oxime carbon–nitrogen double bond (*E*- or *Z*-) in these compounds was so far unknown. In this contribution, we report the results of the determination of the double bond configuration in the solid state by single crystal X-ray diffraction and in solution by 1D and 2D NMR techniques and DFT calculations. It was found that both in the solid state and in solution, **IQ-1** adopts the *E*-configuration stabilized by intermolecular hydrogen bonds, in contrast to previously assumed *Z*-configuration that could be stabilized only by an intramolecular hydrogen bond.

## 1. Introduction

Every year, the number of diseases associated with inflammatory processes in the body, heart dysfunction and the regulation of metabolism is increasing [1]. Enzymes of the c-Jun N-terminal kinase (JNK) family are known to play an important role in human body functioning [2]. JNKs are involved in the regulation of inflammation [3], participate in signaling pathways leading to apoptosis and necrosis [4,5], and regulate some transcriptional and non-transcriptional processes that damage the brain neurons and cardiomyocytes during ischemia/reperfusion [6,7]. JNKs are also involved in the embryonic development of the heart, the regulation of metabolism [8], and the normal functioning of the myocardium. JNK activation is an important link in the pathogenesis of cancer [9], obesity [10], diabetes mellitus [11], insulin resistance [10], and metabolic syndrome [8].

11*H*-Indeno[1,2-*b*]quinoxalin-11-one derivatives are effective and specific inhibitors of the c-Jun N-terminal kinases (JNK) and can be considered as basic compounds for the development of anti-inflammatory drugs [12]. In particular, **IQ-1**, the oxime derivative of 11*H*-indeno[1,2-*b*]quinoxalin-11-one (Figure 1) inhibited the activity of three isoforms of the JNK enzyme and, consequently, the production of proinflammatory cytokines in murine and human monocytic cells [13]. Subsequently, it was shown that **IQ-1** suppresses inflammation and cartilage destruction processes are associated with collagen-induced arthritis [14] and also protect against reperfusion injury in acute cerebral ischemia in mice [15,16,17]. Recently, for sodium salt of **IQ-1** known as IQ-1S, an antihypertensive effect associated with the attenuation of blood viscosity and a decrease in endothelin-1 production was reported [18]. Several derivatives of **IQ-1** with the substituents in the aromatic rings inhibited lipopolysaccharide (LPS)-induced nuclear factor-κB/activating protein 1 (NF-κB/AP-1) activation and interleukin-6 (IL-6) production, which is important for anti-inflammatory activity [19]. Nie et al. demonstrated that IQ-1S protected the mice from sepsis through inhibiting the JNK signaling pathway [20]. The mechanism of IQ-1S anti-inflammatory activity was studied by Seledtsov et al., and it was found that IQ-1S suppresses functionality of both macrophages and T cells [21]. Zhdankina et al. proposed IQ-1S as a promising prophylactic agent for age-related macular degeneration, since it significantly improved the retinal ultrastructure and increased the number of mitochondria in rats [22].

A number of indenoquinoxaline derivatives were synthesized and studied for antitumor activity in [23]. The most active compound was 11-{[3-(dimethylamino)propoxy]imino}-N-[3-(dimethylamino)propyl]-11*H*-indeno[1,2-*b*]quinoxaline-6-carboxamide, which demonstrated antiproliferative activity against cell lines of adenocarcinoma of the breast (MB231), adenocarcinoma of the prostate (PC-3) and hepatocarcinoma (Huh-7). Phosphonate derivatives of **IQ-1** proved to be active against two carcinoma cell lines—human hepatocellular carcinoma (HePG2) and human Caucasian breast adenocarcinoma (MCF7) [24]. Arene–ruthenium complexes with **IQ-1** demonstrated cytotoxicity against pancreatic adenocarcinoma (PANC-1) [25] and cisplatin-resistant breast cancer (MCF7CR) [26] cells.

The natural alkaloid tryptanthrin can be considered as a structural analogue of 11*H*-indeno-[1,2-*b*]-quinoxalin-11-one. Tryptanthrin contains a quinazoline fragment annulated to indole with two carbonyl groups at positions 6 and 12. A large number of natural and synthetic tryptanthrin derivatives containing various substituents at six positions of the indolo[2,1-*b*]quinazoline core are known [27,28,29,30]. Tryptanthrin-6-oxime (**Trp-Ox**, Figure 1) was shown to express JNK3 inhibitory activity comparable to that of **IQ-1** [31]. **Trp-Ox** significantly attenuated the development of collagen-induced arthritis and collagen–antibody-induced arthritis in mice, indicating it as a promising agent against the rheumatoid arthritis [32]. An *O*-acyl derivative of **Trp-Ox** demonstrated strong binding with JNK1-3 isoforms with nanomolar values of dissociation constants and inhibited the production of proinflammatory cytokines, indicating the potential of **Trp-Ox** derivatives as anti-inflammatory drugs [33].

Despite a wide range of potential therapeutic activity, the stereochemistry of the oxime C=N double bond in **IQ-1** and **Trp-Ox** is still unknown; in some works, they were assigned as individual *Z*- or *E*-isomers, or considered as a mixture of dynamically interconverting isomers [34]. However, it is well known that the stereochemical configuration of drugs plays a crucial role in their activity, including the active pharmaceutical ingredients among the class of oximes [35,36]. Thus, fluovoxamine expresses the antidepressant only in the form of *E*-isomer [37]. For better understanding of the mechanisms of action of the oximes **IQ-1** and **Trp-Ox**, it is therefore important to establish the correct stereochemical configurations of these compounds. In this work, we report the experimental, i.e., X-ray diffraction in the solid state and NMR in solution and theoretical investigation of *E*/*Z*-isomerism or **IQ-1** and **Trp-Ox** that allowed to unambiguously assign them as *E*-isomers.

## 2. Materials and Methods

### 2.1. NMR and X-ray Diffraction Equipment

NMR spectra were recorded on a Bruker Advance 500 instrument (Billerica, MA, USA). The solvent residual peaks were used as internal standards. The operating frequencies were 500.13 MHz for ^1^H, 125.76 MHz for ^13^C and 50.69 MHz for ^15^N. The default pulse sequences supplied with the NMR software were used in 2D NMR experiments.

Diffraction data for single crystals of compounds **IQ-1** and **Trp-Ox·Py** were collected with a Bruker D8 Venture diffractometer (Bruker Corporation, Billerica, MA, USA) with a CMOS PHOTON III detector and IμS 3.0 source (mirror optics, λ(CuKα) = 1.54178 Å). The φ- and ω-scanning techniques were employed to measure the intensities.

Diffraction data for single crystals of **Trp-Ox** were obtained on the “Belok/XSA” beamline [38,39] (λ = 0.7527 Å) of the National Research Center “Kurchatov Institute” (Moscow, Russian Federation) using a Rayonix SX165 CCD detector (Rayonix LLC, Evanston, IL, USA). The data were indexed, integrated and scaled, and absorption correction was applied using the XDS program package [40,41].

The structures were solved by the dual-space algorithm (SHELXT [42]) and refined by the full-matrix least-squares technique (SHELXL [43]) in the anisotropic approximation (except hydrogen atoms). Positions of the hydrogen atoms were calculated geometrically and refined in the riding model.

### 2.2. Computational Chemistry Details

#### 2.2.1. NMR Chemical Shift Calculations

The calculations were performed using Gaussian 09 package [44]. The structures of **IQ-1** and **Trp-Ox** were fully optimized at the DFT level of theory employing the three-parameter hybrid B3LYP functional [45,46,47,48] and 6-31+G(d,p) or 6-311+G(2d,p) basis set [49,50,51,52]. The energy minimum character of the found stationary points was confirmed by the absence of imaginary frequencies calculated in a harmonic approximation. Grimme’s empirical dispersion correction with Becke–Johnson damping (GD3BJ) [53] was applied for all B3LYP calculations. Solvation effects were taken into account using the IEFPCM dielectric continuum model [54] and dimethyl sulfoxide as a solvent.

The calculation of the magnetic shielding constants (σ) was performed by the gauge-independent atomic orbital (GIAO) method [55] within the DFT framework using the following functionals and basis sets: WP04 [56]/aug-cc-pVDZ [57]; mPW1PW91 [58]/6-311+G(2d,p) [49,50,51,52]; mPW1LYP [58]/6-311+G(2d,p) [49,50,51,52]; PBE0 [59,60]/6-311+G(2d,p) [49,50,51,52]; mPW1LYP [58]/def2-TZVP [61,62]. The IEFPCM model and the DMSO solvent were used in all the GIAO calculations. The values of chemical shifts relative to TMS (δ, ppm) were calculated by the equation δ = (σ − b)/a; the slope a and intercept b values for the selected computational models were taken from the corresponding publications [63,64].

#### 2.2.2. Thermodynamic Calculations for **IQ-1** and **Trp-Ox**

The calculations were carried out using the ORCA 5.0 software package [65]. The search for energy minima and the calculation of the thermodynamic characteristics of the *E*- and *Z*-isomers of the studied oximes were carried out for the “in” and “out” O-H bond orientations. The D3BJ dispersion correction [53] was taken into account in all the cases of the DFT method application. Model chemistries B3LYP[G] [45,46,47,48,53]/6-31+G(d) [49,50,51,52]/IEFPCM(DMSO) [54] and RI-B2PLYP [66,67]/def2-SVP [61,62]/SMD(DMSO) [68] were used for the geometry optimization of *E*- and *Z*-isomers and transition states for *E*/*Z*-isomerization. Gibbs free energies were calculated using the Quasi-RRHO approach [69] which provides the most correct account for low-frequency normal vibrations, especially important in calculations using the microsolvation model. Nudged Elastic Band with Climbing Image (NEB-CI) methodology [70] was used for finding the transition states of *E*/*Z*-isomerization reactions.

### 2.3. Preparation of Single Crystals for X-ray Diffraction Analysis

#### 2.3.1. 11*H*-Indeno[1,2-*b*]quinoxalin-11-one Oxime (**IQ-1**)

Powder of oxime **IQ-1** was prepared as previously reported [31]. A sample of 10 mg of **IQ-1** was placed in a mixture of 400 µL of acetone and 20 µL of water; the mixture was refluxed on a hot plate until complete dissolution of the solid phase and allowed to cool to room temperature while remaining on a powered-off hot plate.

#### 2.3.2. Tryptanthrin-6-oxime (**Trp-Ox**)

Powder of oxime **Trp-Ox** was prepared as previously reported [31]. A sample of 5 mg **Trp-Ox** was placed in a mixture of 600 µL of dimethylformamide and 420 µL of water; the mixture was refluxed on a hot plate until complete dissolution of the solid phase and then placed in a temperature-controlled oven, in which it was allowed to cool from 100 °C to 30 °C at a rate of 1.4 °C/h (during 50 h).

#### 2.3.3. Tryptanthrin-6-oxime Pyridine Solvate (**Trp-Ox·Py**)

A sample of 18 mg **Trp-Ox** was dissolved in 200 µL of pyridine at 80 °C and allowed to naturally cool to room temperature.

## 3. Results and Discussion

### 3.1. X-ray Crystal Structures

#### 3.1.1. Crystal Structure of 11*H*-Indeno[1,2-*b*]quinoxalin-11-one Oxime (**IQ-1**)

Single crystals of **IQ-1** were obtained by recrystallization from the acetone–water mixture with slow cooling of the solution. The oxime **IQ-1** crystallizes in a monoclinic crystal system, space group *Ia* (Table 1). The asymmetric unit consists of one formula unit (Figure 1) and the unit cell contains four formula units.

Based on the X-ray crystal structure, the configuration of the oxime C=N double bond was unambiguously determined as an *E*-isomer (Figure 1). The geometrical parameters of the oxime group are d(C11–N3) = 1.293(4) Å, d(O1–N3) = 1.386(3) Å, ∠(C11–N3–O1) = 112.3(2)°; the dihedral angle Θ(C11N3O1H1) = 169(4)°.

The molecules are linked into supramolecular chains via hydrogen bonds between the OH oxime group and the nitrogen atom at position 5 of indeno[1,2-*b*]quinoxaline fragment (Figure 2a), d(O1–H1···N1) = 2.731(3) Å, ∠(O1–H1···N1) = 162(5)°. These chains are oriented along the crystallographic axis *c* and the neighboring chains resemble mirror images of each other, resulting in a non-centrosymmetric space group.

The aromatic tetracyclic systems of **IQ-1** are involved in π-π stacking interactions which join them into perpendicular chains. The distance between the planes of the cycles is 3.345 Å (Figure 2b).

#### 3.1.2. Crystal Structure of Tryptanthrin-6-oxime-pyridine Solvate (**Trp-Ox·Py**)

Crystallization of tryptanthrin-6-oxime from pyridine provided single crystals of the solvate **Trp-Ox·Py**. This compound crystallizes in a monoclinic crystal system, space group *P2*_1_/*n* (Table 1). The asymmetric unit is represented by one **Trp-Ox** molecule and one pyridine molecule (Figure 3a); the unit cell contains four formula units. The solvate pyridine molecule is bound to the oxime group of **Trp-Ox** through a hydrogen bond, d(O2–H2···N4) = 2.633(2) Å, ∠(O2–H2···N4) = 176°. The oxime and pyridine molecules are placed almost perpendicular to each other, and the angle between the planes of two molecules is 84.5° (Appendix A). The geometrical configuration of the oxime C=N double bond is *E*- with C2–N3 and N3–O2 bond lengths of 1.293(2) and 1.376(2) Å, and a bond angle ∠(C2–N3–O2) = 110.7(2)°. The dihedral angle Θ(C2N3O2H2) is 178(1)°.

Several types of intermolecular interactions were identified in crystal packing of **Trp-Ox·Py**. Thus, pyridine molecules and the neighboring **Trp-Ox** molecules are involved in C–H···O short contacts via the hydrogen atoms of pyridine rings at position 4 and the carbonyl groups at position 12 of the tryptanthrin tetracycles, d(C18–H18···O1) = 3.347(2) Å, ∠(C18–H18···O1) = 167.7°. These contacts join the molecules into supramolecular chains (Figure 3b). In turn, **Trp-Ox** molecules are involved in π-π stacking which packs the molecules into perpendicular chains (Figure 3c). The distance between the planes of tryptanthrin tetracycles is 3.326 Å.

#### 3.1.3. Crystal Structure of Tryptanthrin-6-oxime (**Trp-Ox**)

In order to avoid the formation of solvate with pyridine, which can make one of the geometrical oxime isomers more favorable and thus influence the geometry of the C=N oxime bond, a series of other solvents (DMSO, DMF, acetone, acetonitrile, their mixtures with water) and crystallization conditions were evaluated to obtain single crystals suitable for X-ray structure determination. It was found that almost in all the tested crystallization conditions, **Trp-Ox** formed amorphous precipitates or microcrystalline conglomerates unsuitable for structural analysis. Only slow cooling of the **Trp-Ox** solution in DMF water afforded a microcrystalline product, the crystal structure of which was determined using synchrotron X-ray radiation.

Tryptanthrin-6-oxime (**Trp-Ox**) crystallizes in a monoclinic crystal system, space group *P2*_1_ (Table 1). The asymmetric unit consists of two formula units of the oxime. One of the **Trp-Ox** molecules in the asymmetric unit is disordered over two equally populated positions differing in the relative orientation of two crystallographically independent molecules (Appendix A). Similarly to the structure of **Trp-Ox·Py**, the oxime C=N bond adopts *E*-configuration (Figure 4). Therefore, formation of pyridine solvate described above is not the main factor determining the C=N bond configuration. The geometrical parameters of the oxime group in **Trp-Ox** are close to those in **Trp-OX·Py**, only the oxime C=N double bond is somewhat longer, d(C11–N3) = 1.319(8) Å, and the N3–O2 bond is slightly shorter (1.347(7) Å), angle ∠(C11–N3–O2) = 113.6(5)°; the dihedral angle Θ(C11N3O2H2) is 180(1)°. The differences in the bond length probably result from different hydrogen bonding networks in **Trp-Ox** and **Trp-Ox·Py**.

The **Trp-Ox** molecules are interconnected by hydrogen bonds involving the oxime OH groups and the nitrogen atoms at position 5 of the tryptanthrin tetracycles (Figure 5a), d(O2–H2···N2) = 2.735(7) Å, ∠(O2–H2···N2) = 161.1°. These hydrogen bonds join the molecules in ladder-type chains oriented along the crystallographic axis *b* (Figure 5b). The chains in the crystal are packed in a spiral fashion, resulting in a chiral space group *P2*_1_.

Summarizing the results obtained from X-ray diffraction analysis, one can conclude that intermolecular hydrogen bonds involving the oxime group can form only for the *E*-isomer. For the Z-isomer, such bonding is sterically hindered and only intramolecular interactions are possible. Apparently, multiple intermolecular interactions involving *E*-oxime molecules are more energetically favorable than single intramolecular hydrogen bonds in the *Z*-isomer, and only the *E*-isomer crystallizes in the solid phase.

### 3.2. Evaluation of the Oxime Group Configuration in ***IQ-1*** and ***Tpr-Ox*** in Solution by NMR Methods

As shown in the previous section, in the solid state, both **IQ-1** and **Trp-Ox** oximes have the double bond *E*-configuration. At the same time, for biomedical applications, it is important to know the spatial structure of the drug in solution.

The solubility of **IQ-1** and **Trp-Ox** in water (about 0.54 mg/L, 2 µM) is insufficient for their study by NMR in water; therefore, dimethyl sulfoxide-d_6_ was used as a solvent.

The ^1^H NMR spectrum of the oxime **IQ-1** contains two signals at 13.36 and 13.26 ppm in the region characteristic of the hydroxyl protons of the oxime group (Appendix A). The ratio of the integral signal intensities is 1.00:0.11. In the resonance region of aromatic protons (7.6–8.6 ppm), the signals of the two forms overlap, but the integral intensity of the signals corresponds to the ratio indicated above (Appendix A). It is obvious that two sets of signals arise from the *E*- and *Z*-isomers of the oxime, with one of them noticeably predominating. In the ^13^C NMR spectrum of compound **IQ-1**, there is a set of 15 signals related to the main isomer, but there are also weak signals that can be attributed to the second isomer (Appendix A).

In the ^1^H and ^13^C NMR spectra of **Trp-Ox**, only one set of signals is observed (Appendix A); therefore, in a solution, this compound is almost completely represented by one isomer. The signal of the oxime proton is located at 13.63 ppm (Appendix A).

The assignment of signals in the ^1^H and ^13^C NMR spectra of **IQ-1** and **Trp-Ox** was performed using 2D NMR spectroscopy methods HSQC (^1^H/^13^C), HMBC (^1^H/^13^C), HMBC (^1^H/^15^N). The 2D NMR spectra plots are shown in Appendix A and the assignment of the signals is given in Table 2.

Nuclear Overhauser effect 2D NMR spectroscopy (NOESY) was used to identify **IQ-1** and **Trp-Ox** isomers. Two possible correlations for the isomeric forms of both oximes are shown in Figure 2. In the experimental NOESY spectra of both **IQ-1** (Appendix A) and **Trp-Ox** (Appendix A), only one cross-peak involving the oxime proton is observed, and in both cases it corresponds to the interactions only possible for *E*-isomers (Figure 2). NOESY interaction of OH protons with H8 (**IQ-1**) or H4 (**Trp-Ox**) in *E*-isomers should not take place because the distance between these atoms (~7 Å) surpasses the upper limit for NOE cross-peak observation (4–5 Å) [71]. In *Z*-isomers, the internuclear distance for OH and H4/H8 pair is in the suitable range (3.3–3.4 Å), but the corresponding cross-peaks are not detected in the NOESY spectra probably due to low concentration of this minor isomer.

To obtain additional confirmation of the oxime double bond geometry, DFT calculations of NMR chemical shifts for *E*- and *Z*-isomers of **IQ-1** and **Trp-Ox** were carried out. Development of computational resources rendered quantum–chemical calculations of NMR spectral parameters a powerful tool for establishing the structures of organic compounds, including complex natural products [72,73,74,75] and even stereoisomers [76,77].

Five computational protocols known for their good accuracy in predicting ^1^H and ^13^C chemical shifts for organic compounds were chosen for the calculations; these protocols are further referred to as Methods A-E. The model chemistries used together with their benchmark root mean square deviations (RSMD) for ^1^H and ^13^C chemical shifts are given in Table 3.

Using the indicated model chemistries, isotropic magnetic shielding constants for *E*- and *Z*-isomers of **IQ-1** and **Trp-Ox** were calculated, and the obtained values were converted to NMR chemical shifts in DMSO using the corresponding slope and intercept coefficients (Table 3). The complete list of the calculated chemical shifts is given in Appendix A and the observed values of RMSD and linear correlation coefficients between the calculated and the experimental chemical shifts (r^2^) are given in Table 4. The obtained prediction accuracy parameters indicate that Method A showed the best performance in terms of RMSD, while Method B provided the best linear correlation between the calculated and the experimental values, which is consistent with the benchmark performance of these computational protocols (Table 3). Representative examples of the correlation plots (obtained for ^1^H chemical shifts using Method A) are shown in Figure 6, and the complete set of the plots is shown in Appendix A.

As one can see from Figure 6 and Table 4, much better calculation–experiment correlations were observed for *E*-isomers of **IQ-1** and **Trp-Ox** for all computational models used. Therefore, a dominating isomer in solution of **IQ-1** as well as the only detected isomer of **Trp-Ox** should be identified as *E*-isomers.

### 3.3. Calculation of Thermodynamic Parameters of the Isomeric ***IQ-1*** and ***Trp-Ox***

Having established the dominating forms of **IQ-1** and **Trp-Ox** in the solid state and in solution, it was necessary to estimate the relative thermodynamic stability of *E*- and *Z*-isomers as well as evaluate the possibility of their interconversion. For this purpose, DFT calculations were carried out for *E*- and *Z*-isomers of **IQ-1** and **Trp-Ox** and the transition state for *E/Z*-isomerization.

Geometry optimization of **IQ-1** and **Trp-Ox** isomers in DMSO using the B3LYP functional, the 6-31+G(d) basis set, and the CPCM solvation model leads to the results presented in Table 5. For both isomers, two possible conformers designated as “in” and “out” with respect to the position of the oxime proton were considered in the calculations.

For both oximes, the *E*-isomer has a thermodynamically more stable “out” conformer, which is obviously explained by a steric repulsion between the hydrogen atoms of the oxime group and the heterocycle in the “in” conformers. *Z*-isomers of **IQ-1** and **Trp-Ox** are more stable in the form of “in”-conformers, probably due to the formation of an intramolecular hydrogen bond OH∙∙∙N.

The search for transition states (TS) of oxime isomerization was carried out using the Nudged Elastic Band with Climbing Image (NEB-CI) methodology [70] followed by refinement of the TS structures. The analysis of normal vibrations shows the presence of a single imaginary frequency, which unambiguously indicates the attainment of the first-order saddle points. The imaginary frequency for both **IQ-1_TS** and **Trp-Ox_TS** corresponds to the in-plane inversion of the oxime nitrogen atom, i.e., the isomerization occurs without rotation around the C=N bond and is not accompanied by breaking of the π-bond. This is consistent with the experimental and theoretical results obtained for other compounds containing the C=N bond [78,79]. In this regard, the geometric structure of the found transition states is characterized by an approximately linear arrangement of the C=N–O atomic group. The calculated barriers for the *E/Z*-isomerization are about 200 kJ/mol (Table 5). Hence, at room temperature in DMSO, the interconversion of the isomers is extremely unlikely.

According to the results presented in Table 5, the *E*-isomers of the studied oximes are thermodynamically more stable than the corresponding *Z*-isomers. Thus, the difference in the Gibbs free energies of two isomers (in the form of the most stable conformers) is 2.05 kJ/mol for **IQ-1** and 0.92 kJ/mol for **Trp-Ox**. However, the use of the double hybrid B2PLYP functional that accounts for the electron correlation in the framework of the second-order perturbation theory, along with the SMD solvation model, more accurate for the solvation energy estimation [68], leads to a greater thermodynamic stability of the *Z*-isomer for both oximes (Table 5). The RI-B2PLYP calculations were carried out by refining the previously optimized structures. In this case, the transition states were optimized using the OptTS option of ORCA 5.0 program. The nature of the refined saddle points remained the same, i.e., the single imaginary frequency corresponded to a planar inversion of the oxime nitrogen atom through an approximately linear configuration of the C=N–O atomic group (Figure 7a,b).

The use of a more accurate approximation does not significantly change the calculated isomerization barriers compared to the data obtained with the B3LYP functional and the CPCM solvation model. However, the calculated stability of **IQ-1** and **Trp-Ox** *Z*-isomers becomes higher than that of the corresponding *E*-isomers by 6–8 kJ/mol (Table 5). It should be noted that the additional thermochemistry calculations for the *E (*out) and *Z (*in) conformers of **IQ-1** at other levels of theory using the SMD model led to similar results. Thus, the values of ΔG°_298_ (*Z*/*E*) and barrier heights (in kJ/mol) are 3.1 and 222 (RI-B2PLYP/def2-TZVP), 3.5 and 217 (B3LYP/6-311++G(2d,2p)), 4.0 and 226 (RI-MP2/def2-SVPD). In all of these cases, the sign of ΔG°_298_ (*Z*/*E*) does not agree with the results of NMR experiments (Section 3.2) indicating a higher stability of *E*-isomers of oximes **IQ-1** and **Trp-Ox**. This discrepancy can be explained by a specific solvation of the oximes by a DMSO molecule which can form an intermolecular hydrogen bond with the oxime OH group in the “out” orientation. To evaluate the effect of such interactions on the relative stability of geometric isomers, DFT calculations of **IQ-1** and **Trp-Ox** were performed using the microsolvation model with one explicit DMSO molecule.

A preliminary search for optimal orientations of one DMSO molecule relative to *E* (out) or *Z* (out) conformers of two studied oximes, as well as the transition states of *Z/E*-isomerization, was carried out using the PBEh-3c composite method [80] that provides high speed and acceptable accuracy of calculations. The effect of bulk solvent was taken into account within the CPCM model. Geometry optimizations of the “oxime**·**DMSO” or “TS**·**DMSO” microsolvates led to an almost linear arrangement of O-H∙∙∙O atom group (Figure 7c,d) with an internuclear distance H∙∙∙O near 1.5 Å indicative of a very strong binding between the highly polar DMSO molecule and the oxime group. The angles H∙∙∙O=S are about 124–125°. It should be noted that according to the Cambridge Structural Database, similar intermolecular interactions between oximes and DMSO solvate molecules are observed in 22 solid-state structures [81,82,83,84,85,86,87,88,89,90,91,92,93,94,95] with the mean O-H∙∙∙O distance 1.816 Å and the shortest distances of 1.65–1.69 Å [88,89].

Based on the optimization results for several structures of each microsolvate differing in the angles of the DMSO molecule rotation about the O-H∙∙∙O axis and the direction of pyramidal sulfur atom inversion, the structures with the lowest energy were selected for further optimization using a double hybrid functional at the RI-B2PLYP/def2-SVP/SMD(DMSO) level of theory. For the *E*- and *Z*-isomers of oximes **IQ-1** and **Trp-Ox** (conformers *E* (out) or *Z* (out)), such structures are characterized by the DMSO methyl group orientation towards the oxime nitrogen atom, whereas in a transition state microsolvate, the sulfur atom of DMSO is directed towards the nitrogen atom of the heterocycle and to the nearest proton of the benzene ring (Figure 7c,d). The vibrational analysis of microsolvates of *E*- and *Z*-isomers confirms the attainment of energy minima on the potential energy surface, while the “TS**·**DMSO” microsolvates have a single imaginary frequency, which again corresponds to the planar inversion of the oxime nitrogen atom. Table 6 shows the Gibbs free energies and some geometric characteristics of the calculated microsolvates.

In microsolvates, the distance r(OH∙∙∙O) for transition states is somewhat shorter than that for the *E*- and *Z*-isomers of compounds **IQ-1** and **Trp-Ox**, in accordance with the larger s-character of the oxime nitrogen atom at the top of the inversion barrier than in the minima on the PES. The corresponding increase in the nitrogen electronegativity can be responsible for the transition state stabilization by the DMSO molecule and for the decrease in the calculated isomerization barriers by about 15 kJ/mol compared with the results obtained using purely continuum solvation models (Table 5; Table 6).

Accounting for the specific solvation with dimethyl sulfoxide leads to higher calculated stabilities of *E*-isomers relative to *Z*-isomers (Table 6), in accordance with the experimental results. Thus, the calculated ΔG°_298_ values for the isomerization process *E*_out_·DMSO → *Z*_out_·DMSO are approximately 11.0 and 14.1 kJ/mol for **IQ-1** and **Trp-Ox**, respectively. The reason for a lower thermodynamic stability of the *Z*-isomer may be its transition from “in” to “out” conformation due to the formation of an intermolecular hydrogen bond with the DMSO molecule and the resulting break of the intramolecular hydrogen bond between the OH group and the nitrogen atom of heterocycle. This is confirmed, for example, by Gibbs energies of the solvate formation in processes (1) and (2) calculated in the RI-B2PLYP/def2-SVP/SMD(DMSO) approximation:*E*_out_ + DMSO → *E*_out_∙DMSO; ΔG°_298_ = −23.6 kJ/mol (**IQ-1**), −25.2 kJ/mol (**Trp-Ox**),(1)
*Z*_in_ + DMSO → *Z*_out_∙DMSO; ΔG°_298_ = −6.5 kJ/mol (**IQ-1**), −3.0 kJ/mol (**Trp-Ox**).(2)

On the other hand, the *E*-isomer, along with the effect of specific solvation, remains stabilized due to the weak intramolecular hydrogen bond between the oxime oxygen atom and the nearest hydrogen atom in the benzene ring: the corresponding distances O∙∙∙H are 2.42 and 2.47 Å in the microsolvates of **IQ-1** and **Trp-Ox**.

We also performed DFT calculations of the microsolvates, each containing one explicit water molecule as a hydrogen bond acceptor attached to the oxime OH group of the transition states and of the *E*- and *Z*-isomer “out” conformers in an aqueous solution. The calculations made in the RI-B2PLYP/def2-SVP/SMD (Water) approximation led to the results very similar to those obtained for the DMSO microsolvates. The calculated ΔG°_298_ values for the *E*_out_·H_2_O → *Z*_out_·H_2_O interconversion equal 6.1 and 12.1 kJ/mol for **IQ-1** and **Trp-Ox**, while the TS·H_2_O microsolvates display Gibbs energies 194.5 and 190.4 kJ/mol above the corresponding *E*_out_·H_2_O species formed by **IQ-1** and **Trp-Ox**, respectively (Figure 8). However, water solubilities of the investigated parent oximes are very low, and efforts were undertaken to synthesize more soluble derivatives like oximates of alkali metals [13,17] and compounds containing amine or carboxylic substituents in the benzene rings [19] of **IQ-1** and **Trp-Ox**.

## 4. Conclusions

In summary, using solid-state X-ray diffraction analysis, 1D and 2D NMR spectroscopy in DMSO solution and DFT calculations it was confirmed that the preferable stereochemistry of the oxime double bond in JNK3 inhibitors **IQ-1** and **Trp-Ox** is *E*-configuration. In a solid state, *E*-isomers are stabilized by intermolecular hydrogen bonds involving the oxime OH group and nitrogen atoms of the heterocyclic rings. In a DMSO solution, according to DFT calculations, the oxime OH group forms a strong hydrogen with the oxygen atom of the solvent molecule, resulting in the stabilization of the E-isomer, while for the *Z*-isomer, formation of the indicated hydrogen bond is sterically hindered. The energy barrier between *Z*- and *E*-isomers in DMSO is about 200 kJ/mol at 298 K, making their interconversion almost impossible at room temperature. Therefore, in biomedical studies involving **IQ-1** and **Trp-Ox** inhibitors, they should be considered as *E*-isomers.

## Data Availability

Experimental data associated with this research are available from the authors. Crystallographic data for the structural analysis were deposited at the Cambridge Crystallographic Data Centre, CCDC No. 2249116 for **Trp-Ox**, 2249117 for **IQ-1** and 2249118 for **Trp-Ox·Py**. Copies of the data can be obtained free of charge from the Cambridge Crystallographic Data Centre, 12 Union Road, Cambridge CB2 1EZ, UK (fax: +44-1223-336-033; e-mail: deposit@ccdc.cam.ac.uk).

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
