# Peer review of "Experimental and Computational Investigation of the Oxime Bond Stereochemistry in c-Jun N-terminal Kinase 3 Inhibitors 11H-Indeno[1,2-b]quinoxalin-11-one Oxime and Tryptanthrin-6-oxime"

_pharmaceutics, 2023, doi:10.3390/pharmaceutics15071802_

Round 1

Reviewer 1 Report

The authors carried out a meticulous study on the stereochemistry of two JNK3 kinase inhibitors, namely IQ-1 and Trp-Ox. They started with X-ray diffraction analyses, followed by NMR spectroscopy with theoretical calculations, and finished off their story with detailed thermodynamic calculations using quantum mechanics approaches. It should be published after minor revisions (and extensive proofreading).

First, I was disappointed when reading that “The solubility of IQ-1 and Trp-Ox oximes in water is insufficient for their study by NMR in water”. Aqueous environment would obviously be much more relevant from a therapeutic perspective than DMSO. Can the authors comment on the exact solubility? This might help future investigations with a higher field NMR spectrometer and/or a larger sample volume. Additionally, the authors may have overreached in their final claim that biomedical studies should consider these two compounds as just E-isomer, considering the very different solvent environments between H2O and DMSO. Have the authors performed similar QM calculations in an aqueous solvent model to lend further support to such a claim?

Second, when discussing NOESY interactions (Scheme 2), the authors should indicate the distance, as predicted/measured in each possible structural model, between each pair of the oxime hydroxyl proton and H1/H8 of IQ-1 or H4/H7 of Trp-Ox. NOE is very distance-dependent and such distance information would greatly help us appreciate these experimental observations.

Author Response

  1. First, I was disappointed when reading that “The solubility of IQ-1 and Trp-Ox oximes in water is insufficient for their study by NMR in water”. Aqueous environment would obviously be much more relevant from a therapeutic perspective than DMSO. Can the authors comment on the exact solubility? This might help future investigations with a higher field NMR spectrometer and/or a larger sample volume. Additionally, the authors may have overreached in their final claim that biomedical studies should consider these two compounds as just E-isomer, considering the very different solvent environments between H2O and DMSO. Have the authors performed similar QM calculations in an aqueous solvent model to lend further support to such a claim?

We have estimated the solubility of both compounds by turbidimetric titration to be about 0.54 mg/L (2 µM), which is below the minimum concentration suitable for recording reliable NMR spectra. Even adding 25 % of DMSO-d6 to D2O did not allow to record the 1H NMR spectra in which the signals would be clearly discernable from the noise. However, submicromolar concentraions of IQ-1 and Trp-Ox were sufficient to express a range of biological activity described in publications cited in the manuscript.

We have added the solubility values to the manuscript to present more information for the reader (Line 261).

Additional DFT calculations were carried out in water using the same microsolvation model and they in fact confirm that the stability of the isomers of IQ-1 and Trp-Ox oximes follows the same trend in water – E-isomer is more stable by about 10 kJ/mol with a large barrier between E and Z isomers (about 190 kJ/mol). Discussion of the stability of isomers in water was added to the text (lines 447-458) and an additional figure with the energy diagram was included (Figure 8).

  1. Second, when discussing NOESY interactions (Scheme 2), the authors should indicate the distance, as predicted/measured in each possible structural model, between each pair of the oxime hydroxyl proton and H1/H8 of IQ-1 or H4/H7 of Trp-Ox. NOE is very distance-dependent and such distance information would greatly help us appreciate these experimental observations.

Scheme 2 was updated and now includes the internuclear distances (from DFT models) for the observed (or hypothetical) correlations. As one can see from the values, the distances for the observed correlations (OH with H1/H7 protons) are very short, while for other possible correlations the distances are to long (7 A) to observe the correlation. The case of 3.3-3.4 A distance is close to the borderline for cross-peak observation (4-5 A) and the corresponding peaks were not detected in the experimental NOESY spectra. The discussion of NOESY was extended (lines 286-292).

Reviewer 2 Report

In this manuscript the authors provide a comprehensive experimental and computational study on the stereochemical configuration of the oxime carbon-nitrogen double of IQ-1 and tryptanthrin-6-oxime, two JNK3 kinase inhibitor.  The methods and data interpretation are accurate and detailed, results are sound and the agreement between theory and experimental data is good.

In my opinion, I believe that this study gives an additional and valued contribution to the field and the paper is suited for publication in this journal.

Author Response

We are thankful to the reviewer for the high estimation of our work.

Reviewer 3 Report

In this manuscript of "Experimental and computational investigation of the oxime bond stereochemistry in JNK3 kinase inhibitors IQ-1 and tryptanthrin-6-oxime", the authors demonstrated the configuration of the oxime carbon-nitrogen double bond in 11H-Indeno[1,2-b]quinoxalin-11-one oxime (IQ-1) and tryptanthrin-6-oxime (Trp-Ox). they employed single crystal X-Ray diffraction in the solid state by single crystal X-Ray diffraction and 1D and 2D NMR techniques in solution, as well as DFT calculations. all the results revealed that both in the solid state and in solution, the E-configuration is more stable by intermolecular hydrogen bonds than the Z-configuration for both of these compounds.

The manuscript is written well, however, it should be minorly revised prior to acceptance for publish in this journal.

1. All the "E-" and "Z-" as configuration should be italic through the manuscript.

2. It is better to present the IQ-1 and Trp-Ox structures in Table 2 with atom numbers.

3. Due to the conclusion that E-isomer is the favor configuration for these two compounds as intermolecular hydrogen bonds, is it possible to expand it to other oxime compounds?

Author Response

1. All the "E-" and "Z-" as configuration should be italic through the manuscript.

Corrected

2. It is better to present the IQ-1 and Trp-Ox structures in Table 2 with atom numbers.

Structures with atom numbers were added to Table 2.

3. Due to the conclusion that E-isomer is the favor configuration for these two compounds as intermolecular hydrogen bonds, is it possible to expand it to other oxime compounds?

Intermolecular interactions are very sensitive to the structure of the compounds, thus expanding the conclusions to other oximes would be highly speculative.